# Biology of *Chrysoperla comanche* (Neuroptera: Chrysopidae)—Should This Predator Be Considered for Insectary Production?

**DOI:** 10.3390/insects16121235

**Published:** 2025-12-06

**Authors:** Kent M. Daane

**Affiliations:** Department of Environmental Science, Policy and Management, University of California, Berkeley, CA 94720-3114, USA; kmdaane@berkeley.edu

**Keywords:** lacewings, insect development, biological control, leafhoppers, *Erasmoneura*

## Abstract

Immature development, larval food consumption, and adult fecundity of *Chrysoperla comanche* (Banks), as a predator of the leafhopper *Erasmoneura variabilis* (Beamer), were determined. Development times for egg, larval stages and pupa were relatively similar to those of closely related *Chrysoperla* species. The larval stages consumed ~250 late instar leafhoppers during a 9.6-day development period, which is comparable to feeding studies for *C. carnea* and *C. rufilabris* feeding on aphids, although prey presentation and stage (size) greatly influence consumption rates. On average, *C. comanche* laid >1000 eggs during a 53.6-day adult lifetime. This study indicated that *C. comanche* is a viable predator of vineyard leafhoppers and may be more suitable for augmentative release in some regions than the commercially available *C. carnea* and *C. rufilabris*. Results are discussed with respect to their mass production by commercial insectaries producing beneficial organisms.

## 1. Introduction

Green lacewings, specifically *Chrysoperla* and *Chrysopa* spp., are important insect predators and biological control agents in horticultural and agricultural ecosystems reviewed in [1,2,3]. Worldwide, *Chrysoperla* species rank as some of the most commonly manipulated and commercially available natural enemies [1,4,5]. However, for many years only two *Chrysoperla* species, *C. carnea* (Stephens, 1836) and *C. rufilabris* (Burmeister, 1839), have dominated commercial insectary programs and field studies in North America and Europe, e.g., [6,7,8,9]. Moreover, the effectiveness of *C. carnea* and *C. rufilabris* augmentative release has varied greatly; for example, studies in North America report from 0 to 100% reduction in targeted pest densities or crop damage [10]. There are numerous factors that can influence lacewing release effectiveness, such as release rate, timing and methodology, which can be controlled by the applicator. Nevertheless, the success or failure of many release programs is contingent upon the biological constraints of the lacewing species released and ecological interactions among lacewings, their prey, and the release environment [1,11,12,13]. Such interactions may be species-dependent, and there are many reasons to assume that *C. carnea* and *C. rufilabris* may not be the most appropriate lacewing species for targeted pests or ecological conditions.

A better understanding of the differences among *Chrysoperla* species will improve biological control programs and highlight the many *Chrysoperla* species, other than *C. carnea* and *C. rufilabris*, that have potential [14,15,16]. For example, *C. carnea* and *C. rufilabris* are similar in most life history traits under humid conditions (>75% RH) but under low to moderate humidity (35–55% RH) *C. carnea* may be more appropriate as *C. rufilabris* has a prolonged pre-ovipositional period, reduced fecundity, increased preimaginal mortality, and a slower developmental rate in hotter and drier conditions [17]. Plant architecture, such as leaf trichomes on cotton or leaf wax levels on cabbage, can also influence *Chrysoperla* spp. performance [10]. Similar to abiotic constraints, there are examples of prey species or prey stage affecting lacewing development and/or survival. *Chrysoperla carnea* was found to be an ineffective predator of silverleaf whitefly, *Bemisia tabaci* (Gennadius, 1889), and the greenhouse whitefly, *Trialeurodes vaporariorum* (Westwood, 1856), in part, because the predator’s nutritional demands were only marginally met by feeding on whiteflies [18,19].

Here, we report on key biological parameters of *Chrysoperla comanche* (Banks, 1938) as a potential predator for use in biological control programs. We became aware of *C. comanche*’s prevalence in agricultural systems while studying commercial augmentation of *C. carnea* in California vineyards to suppress two leafhopper pests, *Erasmoneura variabilis* (Beamer, 1929), formerly *Erythroneura* [20], and *E*. *elegantula* Osborn, 1928 [21,22]. In post-release surveys, five green lacewing species were found: *C. carnea, C. comanche*, *Chrysopa coloradensis* Banks, 1895, *Chrysopa nigricornis* Burmeister, 1839, and *Chrysopa oculata* Say, 1839. The *Chrysopa* spp. were most commonly found on the ground covers but rarely on the vines, *C. carnea* was found on both the vines and the ground covers, and *C. comanche* was the most commonly found species on the vines. These results suggest that *C. comanche* may prove to be a better biological control agent in California San Joaquin Valley vineyard ecosystems. Nevertheless, *C. comanche* and other green lacewing species have received less attention as a manipulated generalist predator, and, for this reason, there are fewer studies on its biology and ecology. The study objectives were to determine *C. comanche* egg, larval and pupal development times; larval food consumption; and adult fecundity. The data provide some of the basic information needed to evaluate *C. comanche* as a potential biological control agent in comparison with *C. carnea* and *C. rufilabris*.

## 2. Materials and Methods

### 2.1. Insect Material

A laboratory colony of *C. comanche* was established with field-collected adults from vineyards in Fresno County, CA, USA. Lacewings were identified using a key developed by Dr. Hagen, University of California, Berkeley. Adults were fed an artificial diet made of whey yeast, an enzymatic protein hydrolysate of Brewer’s yeast, honey, and distilled water (6:1:10:5), using the procedures described by Johnson and Hagen [23]. Lacewing larvae were fed eggs of the Mediterranean flour moth, *Ephestia kuehniella* (Zeller, 1879), that were produced on a wheat bran mixture in 3.8 L glass jars topped with a cotton cloth as an oviposition substrate. The collected eggs were either used to maintain the moth colony or frozen and used as lacewing food. Lacewings used in the experiments were within F_2–6_ generations of field-collected material. Cultures were held at 25 ± 1 °C, 60 ± 10% RH and 16:8 photoperiod. Leafhoppers (*E. variabilis*) used in the prey-consumption study were field-collected on vines at the Kearney Agricultural Research and Extension Center (KREC) and used within 24 h of collection.

### 2.2. Temperature Dependent Development

The effect of constant rearing temperatures on *C. comanche* development time was tested at 12.7, 15.6, 21.1, 26.7, 29.4, 32.2, 33.9, 35 and 36.7 °C. To collect fresh eggs, adult female *C. comanche* were isolated in three 3.8 L cylindrical, waxed cardboard cartons, with ~20 adults per carton. The cartons contained strips of wax paper, which were provided as an oviposition substrate. After 4 h, eggs were harvested, isolated singly in 55 mL glass vials, and randomly assigned to a temperature treatment. Before the neonate larvae hatched, the vials were stocked with fresh *E. kuehniella* eggs to provide an immediately available food source. Thereafter, the vials were resupplied with *E. kuehniella* eggs every 2–3 days, providing an overabundance of food. Lacewing development was checked daily, recording developmental stages as: egg, first, second or third instar, pupa, or adult. There were 20–42 *C. comanche* tested at each temperature. After pupation, 3-day-old cocoons were weighed to estimate the influence of temperature on lacewing growth. Temperatures (T) were maintained at T ± 1 °C; 60 ± 10% RH, with a 16:8 (L:D) photoperiod; due to the availability of temperature cabinets, treatments at 15.6, 21.1, 26.7, and 32.2 °C were first tested, and then treatments 12.7, 29.4, 33.9, 35 and 36.7 °C were tested.

### 2.3. Adult Fecundity, Longevity and Overwintering

*Chrysoperla comanche* adult fecundity was tested using adults reared from field-collected eggs in KREC vineyards. After hatching, neonate larvae were provided with an excess of *E. kuehniella* eggs and larvae throughout lacewings’ larval stages. Upon adult emergence, lacewings were sexed, and male and female pairs were confined individually in 70 mL containers, ventilated by a 3 cm organdy-covered hole. A strip of wax paper placed under the lid served as an oviposition substrate. The adults were provided with an ad lib supply of artificial diet, as described above, and water was provided using a water-saturated cotton ball. Male lacewings were removed from the containers 7 days after egg deposition was first observed, and females were transferred to new vials every 2 days throughout their lives. The numbers of eggs deposited were recorded. There were 20 adults tested. The trial was conducted at 26.7 ± 1 °C, 60 ± 10% RH, and a photoperiod of 16L:8D.

*Chrysoperla comanche* overwintering viability at ambient temperatures in the San Joaquin Valley was assessed by holding field-collected adults in plastic containers (3.3 cm × 9.5 cm × 9.5 cm) in an outdoor sleeve cage from 26 October to 7 January. The adults were maintained on the same artificial diet, described above, and were transferred to new containers twice a week (Monday and Friday). *Chrysoperla comanche* incubation period and egg production were recorded haphazardly throughout the trial period.

### 2.4. Prey Consumption

The potential number of *E. variabilis* consumed by *C. comanche* during the larval period was determined. Lacewing eggs were confined singly in 35 mL glass vials with an organdy cover for ventilation. Upon hatching, larvae were supplied with freshly collected 4th or 5th instar leafhoppers every 2 days during the lacewings’ 1st and 2nd instar stages (25–50 leafhoppers each period) and daily during the 3rd instar (50–75 leafhoppers daily). The leafhoppers were presented on a grape leaf bouquet by cutting a 1.5 cm × 8 cm leaf section by the midrib and placing the remaining petiole section into a 5 mm vial filled with tap water and held in place by cotton around the petiole. The consumed (killed) leafhoppers were identified by their dried and shrunken bodies, and their numbers were daily recorded (initial work with a no-lacewing control established that there was no *E. variabilis* mortality on the bouquets). After pupation, 3-day-old cocoons were weighed. There were 20 adults tested. The trial was conducted at 26.7 ± 1 °C, 60 ± 10% RH, and a photoperiod of 16L:8D.

### 2.5. Statistics

Results are presented as sample means ± SE. Analyses were performed using Systat Software Inc. (version 13, San Jose, CA, USA). A nonlinear regression analysis was used to describe the relationship between temperature and *C. comanche* developmental rate (egg to adult eclosion), using the Brière-2 equation [24,25]:R(*T*) = α *T* (*T* – *T*_L_) (*T*_H_ − *T*)^(1/*m*)^
where *T* is the rearing temperature (°C), α is a constant fitted to the data, *T*_L_ is low temperature development threshold, *T*_H_ is the high temperature threshold, and *m* is an empirical constant. To better determine the low temperature, data within the mid-range temperature treatments (15.6–32.2 °C) were fit to a linear equation [25], as nonlinear models often provide a poor fit to T_L_:R(*T*) = *a* + *b*T
where the development rate R(*T*) is a linear function of temperature, T(°C), *a* is the intercept of the line and *b* is the slope of the regression line. The low development threshold is calculated as *T*_L_ = −*a*/*b*, and the thermal constant (*k*) from birth to adult, in required degree-days (DD), is calculated as *k* = 1/*b* [26]. The optimal temperature was then computed as follows [26]:*T*_opt_ = (2*m T*_H_ + (*m* + 1) *T*_L_) + ((4*m*^2^
*T*_H_^2^ (*m* + 1)^2^ *T*_L_^2^ − 4*m*^2^
*T*_H_
*T*_L_))^0.5^/4*m* + 2

This expression depends only on T_H_, T_L_, and *m*, which were previously determined.

## 3. Results

### 3.1. Temperature-Dependent Development

Tested *C. comanche* completed development at temperatures from 15.6 to 35 °C; eggs did not hatch after 3 weeks, at the lowest (12.7 °C) or highest (36.7 °C) temperatures tested. Therefore, the temperatures tested covered the range of constant temperatures that permit complete development. From egg to adult eclosion, developmental time was longest at 15.6 °C (56.1 ± 2.9 d) and shortest at 32.2 °C (16.7 ± 0.7 d), with development times relatively similar between 26.6 and 29.4 °C and 32.2 and 33.9 °C (Table 1). At all temperatures, development times of all instars (1st to 3rd) were relatively similar to pupal development time, and eggs eclosed in about half that time.

The linear model, using the mid-range temperatures from 15.6 to 32.2 °C, provided low temperature development rates for egg (10.7 °C, y = −0.167 + 0.015x, R^2^ = 0.97), first instar (9.7 °C, y = −0.195 + 0.020x, R^2^ = 0.97), second instar (8.7 °C, y = −0.198 + 0.023x, R^2^ = 0.95), third instar (8.5 °C, y = −0.179 + 0.021x, R^2^ = 0.92), and pupa (8.0 °C, y = −0.045 + 0.005x, R^2^ = 0.92). From egg to adult eclosion, the low temperature threshold was 9.97 °C (y = −0.017 + 0.002x, R^2^ = 0.97). The corresponding accumulated degree days are 64.4, 49.7, 44.1, 47.8, and 173.2 for egg and first, second, third instar and pupa, respectively; the accumulated degree day from egg to adult emergence is 519.4. The proportion of time spent in each stage was relatively similar for each temperature (15.6–35 °C), ranging from 15.2 to 21.5% (egg), 36.5–44.3% (all instars) and 39.9–46.8 (pupa).

Using these low temperature thresholds and setting the upper threshold at 36 °C (there was complete development at 35 °C and no survival at 36.7 °C), the Briere-2 nonlinear model provided a good fit to the data set for development for egg, larvae, pupa and egg to adult eclosion (Figure 1) and provides T_opt_ as 25.8, 28.2, 28.9, 28.8, 31.6, and 29.7 °C for egg, fist instar, second instar, third instar, pupa and egg to adult eclosion, respectively.

Weight of 3-day-old pupae ranged from 11.5 ± 1.8 mg at 15.6 °C to 5.9 ± 0.25 mg at 35 °C. Weight was negatively associated with increasing temperature using a linear model (y = 16.37 − 0.27x; R^2^ = 0.84, F = 33.19, df = 1.6, *p* = 0.002; Figure 2, dashed line). A simple quadratic equation better captured the decline in weight at both lower and upper temperatures to show a negative relationship with a sharp decline as temperatures reached the lethal limits (y = 5.78 + 0.629x − 0.018x^2^; R^2^ = 0.96, F = 1417.8, df = 3.7, *p* < 0.001; Figure 2, solid line).

### 3.2. Adult Fecundity and Longevity

Adults had an average of 5.8 ± 0.9-day pre-ovipositional period and deposited an average of 1087.2 ± 199.5 eggs over the lifetime of 53.6 ± 10.2 days (Figure 3). Egg production reached a peak in about 10 days and steadily declined thereafter, with 80.9% of eggs deposited in the first 30 days of oviposition.

Overwintered adults continuously deposited eggs from 26 October to 7 January. Eggs deposited from 16 to 19 November hatched from 7 to 13 January after 49–53 days at an average temperature of 7.8 °C (range −7.7 to 26.7 °C). Adult *C. comanche* were collected from grape vines and prune trees at KREC in mid-November, when all or most leaves had senesced and dropped, and collected *C. comanche* adults were green in color (compared to the brown color of diapausing *C. carnea*). Therefore, it appears that *C. comanche* does not enter a diapause in winter in California’s San Joaquin Valley.

### 3.3. Prey Consumption

Larvae consumed or killed 252.6 ± 9.2 late-instar *E. variabilis* in an average 9.6 ± 0.2-day larval development period. More leafhoppers were consumed in the third instar (200.2 ± 8.6) than in the second instar (41.4 ± 3.0), which consumed more than the first instar (10.9 ± 1.2) (F = 371.02, DF = 2.39, *p* < 0.001; Figure 4).

## 4. Discussion

Two *Chrysoperla* species dominate the commercial insectary market—*C. carnea* and *C. rufilabris* in North America [27] and *C. carnea* in Europe [5,28]. Other species have been occasionally available, such as *C. externa* and *C. nipponensis* (Okamoto, 1914) in Latin America and Asia [29,30]. Still, there have been many studies indicating that other lacewing species are better suited to some environments or against some targeted pests—based on the presence of different species, e.g., [14,31]. For example, the brown lacewing *Micromus angulatus* (Stephens, 1836) and *Chrysopa formosa* (Brauer, 1851) were shown to control the green peach aphid *Myzus persicae* (Sulzer, 1776) on peppers, and the authors highlight the potential of these two widespread but overlooked species for use in biological pest control [16]. Cortez-Mondaca et al. [15] showed that *Ceraeochrysa cubana* (Hagen, 1861) and *C. externa* were abundant in sorghum fields infested with sugarcane aphid, *Melanaphis sacchari* (Zehntner, 1897) in Mexico and suggested the mass rearing and release of *C. cubana*. Finally, Koutsoula et al. [14] showed that *C. agilis* (in the *carnea* species group) and *Chrysoperla mutata* (McLachlan, 1898) (in the *pudica* species group) were efficient in controlling aphids and mealybugs on sweet pepper and suggested their use in pest control programs. This study focused on *C. comanche*, which is considered to be native to western North America. The natural range of *C. comanche* appears to be better suited to warm-hot, dry habitats, and there are numerous reports of *C. comanche*’s importance in agricultural systems within this region. For example, *C. comanche* and *C. nigricornis* were abundant in western US pecan trees feeding on two specialist aphids, *Monellia caryella* (Fitch, 1855) and *Melanocallis caryaefoliae* (Davis, 1910) [32]. In a glasshouse study in Mexico, *C. comanche*, along with *C. externa*, was shown to reduce tomato fruit damage from *Frankliniella occidentalis* (Pergande, 1895) [33]. Here, the development, larval food consumption and adult fecundity of *C. comanche* were determined to better compare this lacewing species to *C. carnea* and *C. rufilabris* that are most commonly used in augmentation programs.

The developmental time for *C. comanche* was 15.4 days at 26.7 °C from neonate first instar to adult eclosure (Table 1). The proportion of total developmental time that *C. comanche* spent in the egg, larval and pupal stages was relatively similar among temperatures, suggesting rate isomorphy [34]. In comparison, Peterson and Hunter [32] report a slightly faster time of 16.4 days (females) and 16.0 days (males) at 27 °C, the only other developmental study found for *C. comanche*. In their study, *C. comanche* was fed the pecan aphids *M. caryella* and *M. caryaefoliae*, which may have slowed development time because of the additional energy used and handling time for prey capture. The cocoon weight (9.9 mg) reported herein was also larger than the 9.5 mg (females) and 7.7 mg (males) reported [32], but this might have also been a result of different diets and the fact that female and male specimens were not weighed separately. Many factors can impact development time, making comparisons of different studies difficult. For example, *C. carnea* that were resistant to Spinosad had a shorter development time than a susceptible population [35], *C. carnea* fed increasing prey (aphids) densities had reduced developmental time [36,37], and *C. rufilabris* fed pea aphid, *Acyrthosiphon pisum* (Harris, 1776), reared on alfalfa developed faster than those fed pea aphids reared on faba beans [38]. Even in this study, the developmental time of larvae fed leafhopper nymphs was a day longer compared to larvae fed moth eggs held at the same temperature.

Work conducted on *C. carnea*, *C. rufilabris* and other *Chrysoperla* spp. suggests that developmental times are, within reason, similar to *C. comanche*. For example, Amarasekare and Shearer [39] report that *C. carnea* and *C. johnsoni* Henry et al., 1993 developed from egg to adult, at 23 °C, in 28.5 and 31.8 days, respectively. Giles et al. [38] report that *C. rufilabris* developed from egg to adult, at 22 °C, in 22 and 24.1 days when fed pea aphids reared on alfalfa and faba beans, respectively. These development times are comparable to *C. comanche* developmental times of 33.6 and 19.5 days, at 21.1 and 26.7 °C, respectively (Table 1). Fewer studies used a range of constant temperatures to determine lower and upper thresholds. Aghdam and Nemati [40] investigated the *C. carnea* developmental rate at seven constant temperatures and analyzed that data with nonlinear models. Their reported lower and upper temperature thresholds were higher and lower, respectively, at each developmental stage for *C. carnea* than those reported here for *C. comanche*. Similarly, Butler and Ritchie [41] report that *C. carnea* upper threshold temperatures were about 2 °C lower, for each life stage, than *C. comanche* reported here. Thus, *C. comanche* is probably better adapted to warmer than cooler regions as compared with *C. carnea*. The estimated upper threshold for *C. comanche* (36.5 °C) is not outside of that reported for other chrysopids, with Pappas et al. [42] reporting *Chrysoperla agilis*. Henry et al. 2003 reported a European chrysopid in the *C. carnea* group, with a lower threshold between 11.4 and 11.8 °C and an upper threshold between 36.6 and 36.9 °C for females and males, respectively.

Weight of 3-day-old *C. comanche* pupae was negatively associated with increasing temperature, with a sharper decline near the upper temperature threshold. For many species, the temperature-size rule suggests that insects reared at cooler temperatures develop more slowly but reach a larger final body size and mass than those raised in warmer conditions [43]. *Chrysoperla comanche* larvae consumed or killed >250 late-instar *E. variabilis* in about a 10-day larval development period. Based on the size difference between first instar leafhoppers and the late instars used in the trial, the *C. comanche* larvae are probably capable of consuming >1000 early instar leafhoppers. Under field conditions, larvae should be expected to consume fewer prey and will probably require a longer development time, because of additional searching time and energy consumption used to find prey. No other studies detail *Chrysoperla* spp. consumption of leafhoppers; however, Luna-Espinosa et al. [33] investigated predation of adult *F. occidentalis* on tomato by *C. comanche* and *C. externa* larvae and showed relatively similar numbers of 258 and 297 prey killed, respectively. Still, direct comparison is difficult as the researchers used a 24 h period for each lacewing stage, thus reducing the total lifetime consumption, but they did show a similar trend of greater consumption by the later lacewing instars.

Many feeding studies have used moth eggs or aphids as prey, with results commonly showing that *C. carnea* or *C. rufilabris* will consume 100s to 1000s of moth eggs, e.g., [37,44,45,46,47,48] or 100s of aphids, e.g., [47,49,50,51]. Basically, most *Chrysoperla* spp. are generalist predators of soft-bodied insects and mites, a trait that underlies their great commercial demand. Still, *Chrysoperla* spp. prey preferences can vary [47,52,53] and could be better defined among species to provide reliable recommendations for the improved use of *Chrysoperla* species and biotypes against specific types of pests [1,54]. Moreover, laboratory prey consumption studies indicate that lacewings can feed and develop on specific prey species, but their effectiveness in the field can be quite different depending on conditions at the release site [55]. As examples, smooth or hirsute cotton leaves affect *C. rufilabris* larval mobility and prey consumption [56], and the structure of cabbage and wheat plants impacts the effectiveness of *C. carnea* [49,57]. Even more subtle differences can occur—in pecan, *C. rufilabris* can feed on the aphids *M. caryella*, *M. caryaefoliae*, and *Monelliopsis pecanis* Bissell, 1983, but will lay more eggs on trees infested with the *M. caryella* [58].

*Chrysoperla comanche* oviposited >1000 eggs over about 55 days, with about a 6-day pre-ovipositional period. While the recorded fecundity may seem high, it is actually in line with other studies that used well-fed larvae and an artificial diet to increase adult longevity. Amarasekare and Shearer [39] report that *C. carnea* and *C. johnsoni* had a lifetime fecundity of 1264 and 960 eggs per female, respectively. Greenberg et al. [59] reported *C. rufilabris* with 440.6 to 802.6 eggs per female, depending on larval prey species. What is clear is that adult *Chrysoperla* spp. fecundity is dependent on larval diet, with Zheng et al. [37] reporting 217 to 1205 eggs per female from low to high prey densities of *E. kuehniella* moth eggs and Athhan et al. [36] reporting 514.2 to 806.6 eggs per female fecundity from low to high densities of *Hyalopterus pruni* (Geoffer, 1762) aphid nymphs. Adult females that developed from the high feeding treatment not only had higher fecundity but also a substantially shorter preoviposition period and a later decline in egg deposition as well [37].

## 5. Conclusions

Here, the development, larval food consumption and adult fecundity of *C. comanche* were compared with the commercially available *C. carnea* and *C. rufilabris*. At the onset of this study, the objective was to determine if *C. comanche* might be a better fit for augmentative release in perennial crops in California’s interior valleys. Clearly, there are differences among lacewing species; for example, *C. carnea* does well under low humidity and could be released in hot, dry agricultural settings, whereas *C. rufilabris* does better under high humidity and might be best used in greenhouses rather than the desert southwest of the USA [1]. Here, we showed that, while most biological traits were similar, *C. comanche* had a higher temperature threshold, and adults appeared not to enter a winter diapause in California’s San Joaquin Valley. Still, a general call for greater species diversity among commercially available *Chrysoperla* spp. may not be feasible. At the insectary level, first, there is now a more global market for beneficial insects, and import/export permit regulations for international shipment of an even greater number of beneficial insects would be impractical. Second, mass rearing of closely related species at one facility could be problematic because of contamination of insect colonies, and that would then require more costly examination of product for species identification before shipments. Third, product loyalty for the different generalist natural enemies would be disrupted by consumers trying different lacewing species without a clear understanding of each species best usage practices, whereas *C. carnea* and *C. rufilabris* have been known commodities for some time.

At the field level, it has been abundantly clear that release rates, release methods and release environment probably have more impact on *Chrysoperla* than differences among species or biotypes. For example, lacewing releases reduced aphid abundance in apples, but release efficacy is dependent on release method, life stage, and timing [60]. There is also the potential impact of intraguild predation that could release efficiency [61,62,63]. For example, the Argentine ant, *Linepithema humile* (Mayr, 1868), removed 98% of the *C. carnea* eggs that were dispensed on tulip trees to control the aphid *Illinoia liriodendri* (Monell, 1879) [64]. Therefore, while the data presented here suggest that *C. comanche* has potential as a biological control agent of *E. variabilis*, it may be unreasonable to call for the larger commercial insectaries to offer *C. comanche* as a mass-produced beneficial arthropod—or any of the other lacewing species discussed. Rather, specialty crop systems and beneficial arthropods that are especially advantageous for specific situations might best be produced by cooperatives or larger farm operations that address specific pest needs that cannot be met by one of the hundreds of beneficial arthropods currently under commercial production.

## Figures and Tables

**Figure 1 insects-16-01235-f001:**
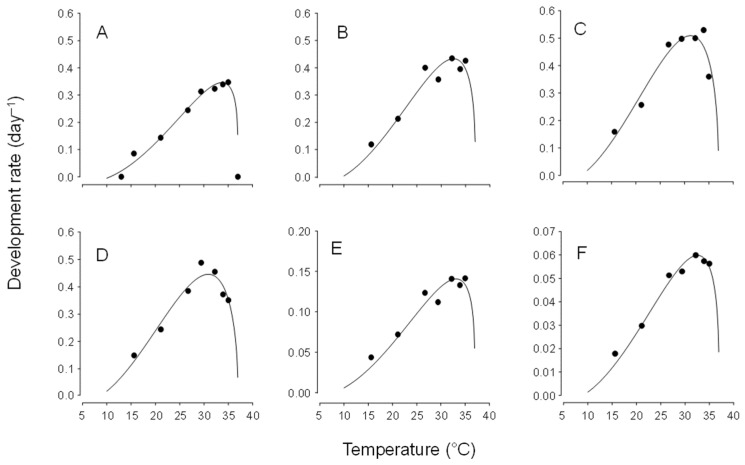
The relationship between temperature and development rate of *Chrysoperla comanche* at constant temperatures, as described by the Brière-2 nonlinear model [24] for (**A**) egg (R^2^ = 0.97), (**B**) first instar (R^2^ = 0.97), (**C**) second instar (R^2^ = 0.95), (**D**) third instar (R^2^ = 0.92), (**E**) pupa (R^2^ = 0.92) and (**F**) egg to adult eclosion (R^2^ = 0.97).

**Figure 2 insects-16-01235-f002:**
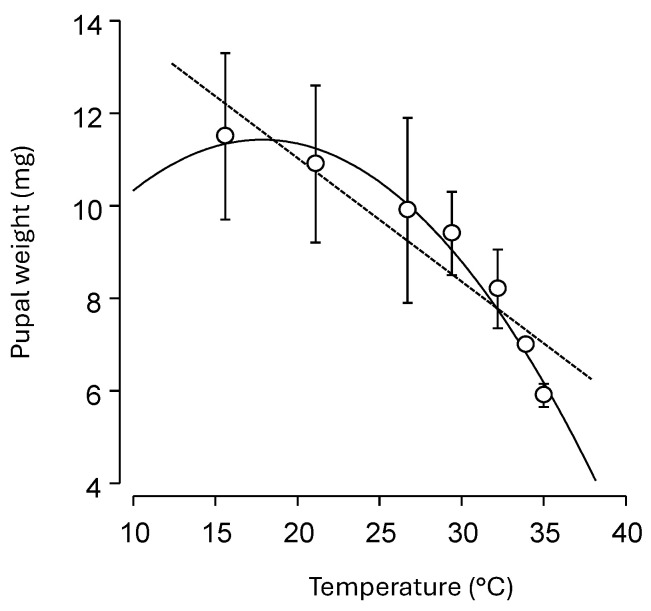
Relationship between constant temperature and average (±SEM) weight of 3-day-old cocoons of *Chrysoperla comanche*; dashed line fits data to a linear model, solid curve fits data to a simple quadratic model.

**Figure 3 insects-16-01235-f003:**
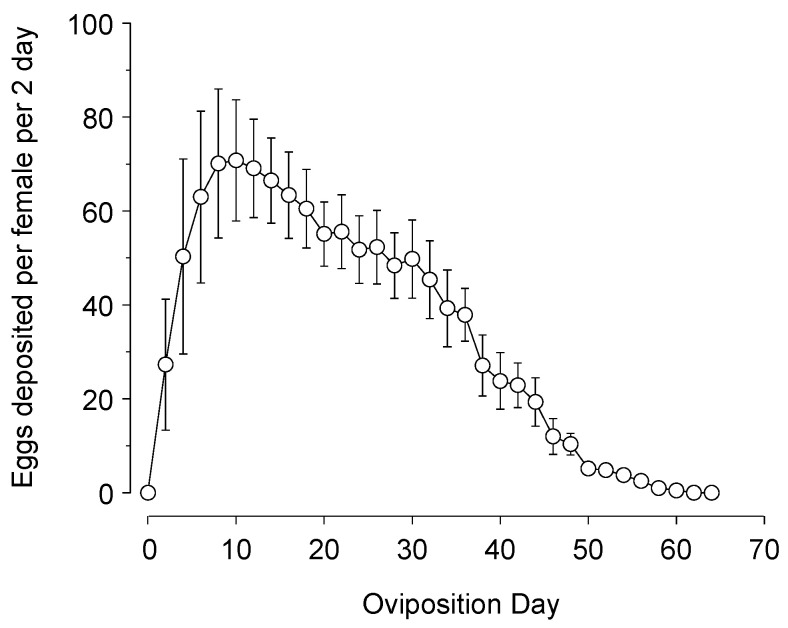
Lifetime fecundity of *Chrysoperla comanche* (eggs per 2 days ± SEM) when fed an artificial diet made of whey yeast, enzymatic protein hydrolysate of brewer’s yeast, honey and water (6:1:10:5).

**Figure 4 insects-16-01235-f004:**
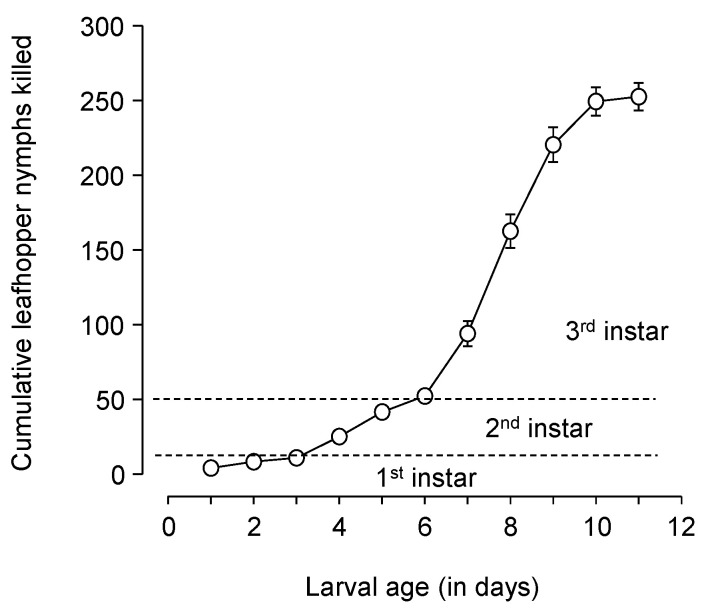
*Chrysoperla comanche* cumulative consumption of late instar (4th and 5th instar) leafhopper nymphs, in a controlled environment with prey provided in excess, showing more consumption in the lacewing’s 3rd instar than the 1st and 2nd instars combined. The dashed lines show the approximate time periods for the three instars at 26.7 °C.

**Table 1 insects-16-01235-t001:** Development time (in days ± SEM) for different life stages of *Chrysoperla comanche* at seven constant temperatures that permitted complete development.

Life Stage	T (°C)						
	15.6	21.1	26.7	29.4	32.2	33.9	35.0
Egg	11.8 ± 0.4	7 ± 0	4.1 ± 0.4	2.9 ± 0.1	3.6 ± 0.5	2.9 ± 0.1	2.9± 0.4
1st instar	8.4 ± 1.9	4.7 ± 0.5	2.5 ± 0.5	3.0 ± 0.1	2.0 ± 0	2.5 ± 0.1	2.3 ± 0.1
2nd instar	6.3 ± 0.5	3.9 ± 0.3	2.1 ± 0.4	2.0 ± 0.0	2.0 ± 0	1.9 ± 0.1	2.8 ± 0.1
3rd instar	6.7 ± 0.9	4.1 ± 0.3	2.6 ± 0.5	2.1 ± 0.0	2.1 ± 0	2.7 ± 0.2	2.8 ± 0.1
All instars	21.4 ± 1.7	12.7 ± 0.5	7.3 ± 0.5	7.1 ± 0.1	6.1 ± 0.01	7.0 ± 0.2	7.9 ± 0.1
Pupa	22.9 ± 1.1	13.9 ± 0.8	8.1 ± 0.4	8.9 ± 0.0	7.1 ± 0.3	7.5 ± 0.1	7.1 ± 0.1
Egg-adult	56.1 ± 2.9	33.6 ± 0.7	19.5 ± 0.5	19.0 ± 0.1	16.7 ± 0.7	17.4 ± 0.1	17.8 ± 0.1

## Data Availability

The original contributions presented in this study are included in the article. Further inquiries can be directed to the corresponding author.

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
