# Peer review of "Biology of *Chrysoperla comanche* (Neuroptera: Chrysopidae)—Should This Predator Be Considered for Insectary Production?"

_insects, 2025, doi:10.3390/insects16121235_

Round 1

Reviewer 1 Report

Comments and Suggestions for Authors

The paper by Kent M. Daane deals with the life history characteristics of a green lacewing species, which can be used as commercial predatory insects. Knowing details about the biology of a commercial insect species always has scientific merit, even if it is in a certain geographic region. Methodology is sound and results well interpreted. The discussion part is also sound interpreting results in the context of biological control. I have a few specific and one general comment:

- To have sufficient information in the title please add (Neuroptera: Chrysopidae) in the title after the name Chrysoperla comanche.

- One comment for the Material and Methods section: Since C. comanche and C. rufilabris have the ability to hybridize and cannot be distinguished based on mtCOI (see Gebiola & Stouthamer, Journal of Economic Entomology, 2019), please elaborate on how the colony has been established and confirmed as C. comanche.   

- Comments on several nomenclature issues. The name Erythroneura variabilis Beamer, 1929 is no longer valid. The accepted and preferred name of the species is Erasmoneura variabilis (Beamer, 1929), see reference - Dietrich, C.H. & Dmitriev, D.A. (2006) Review of the New World genera of the leafhopper tribe Erythroneurini (Hemiptera: Cicadellidae: Typhlocybinae). Bulletin of the Illinois Natural History Survey, 37(5), I-IV, 119–190. Please change the name of the species accordingly.

- Please add the original author and date of each species mentioned in the manuscript, first time it is mentioned.

- Please check throughout the manuscript, whether organism names are written correctly, e.g., the author of Chrysoperla comanche is written in the manuscript both with and without the parentheses. Please check that all species names mentioned in the text are written properly.

Author Response

Comment 1 - "To have sufficient information in the title please add (Neuroptera: Chrysopidae) in the title after the name Chrysoperla comanche." Response 1 - I have added "(Neuroptera: Chrysopidae) and shortened the title to "Biology of Chrysoperla comanche (Neuroptera: Chrysopidae) – should this predator be considered for insectary production?

Comment 2 - "One comment for the Material and Methods section: Since C. comanche and C. rufilabris have the ability to hybridize and cannot be distinguished based on mtCOI (see Gebiola & Stouthamer, Journal of Economic Entomology, 2019), please elaborate on how the colony has been established and confirmed as C. comanche" Response 2 - We used a key developed by Ken Hagen to separate the Chrysoperla spp. and did not have a colony of C. rufilabris in the lab (at that time we purchased C. rufilabris from insectaries, but it was never in the rearing room).

Comments 3-5 (insect nomenclature) 

Comments on several nomenclature issues. The name Erythroneura variabilis Beamer, 1929 is no longer valid. The accepted and preferred name of the species is Erasmoneura variabilis (Beamer, 1929), see reference - Dietrich, C.H. & Dmitriev, D.A. (2006) Review of the New World genera of the leafhopper tribe Erythroneurini (Hemiptera: Cicadellidae: Typhlocybinae). Bulletin of the Illinois Natural History Survey, 37(5), I-IV, 119–190. Please change the name of the species accordingly. Please add the original author and date of each species mentioned in the manuscript, first time it is mentioned. Please check throughout the manuscript, whether organism names are written correctly, e.g., the author of Chrysoperla comanche is written in the manuscript both with and without the parentheses. Please check that all species names mentioned in the text are written properly. Response 2 - I have changed the genus name and included the citation above. I have checked all of the species nomenclature and made corrections as needed.

Reviewer 2 Report

Comments and Suggestions for Authors

The study examines parameters such as larval development, prey consumption, and fertility of Chrysoperla comanche. Materials and methods are explained in detail and the results are well represented with the aid of graphs. The discussion provides an accurate comparison between the characteristics of C. comanche and those of the commercially available Chrysoperla species, highliting that each Chrysoperla species is better suited to some environments or against some pests. The article conclusions are that since release conditions influence Chrysoperla impact more than differences among species or biotypes and since it is not feasible the insectary mass producing of species such as C. comanche, it would be more realistic its production by cooperatives that address specific pest needs.

The study is accurate and, above all, sincere and realistic in its conclusions.

Author Response

Comments 1 - the reviewer praised the manuscript, but has no specific suggestions. Response - thank you.

Reviewer 3 Report

Comments and Suggestions for Authors

This study on the parameters of a "new" beneficial predator is very well done. All parts = introduction, methods, results, and mainly discussion are well-written. Only the conclusions chapter is too long and rather based on literature than on own discoveries. 

Still, I have several suggestions to improve the article, more described in the pdf attached:

143: Because of the functional response rules, you have to determine roughly the number of prey provided.

186: Can you summarize lower thresholds and optimal temperatures in a table together with standard errors?

206: Find a suitable non-linear equation for body mass.

Author Response

Attached file: Response: Thank you for taking the time to add suggestions to the manuscript. I have made all of the suggested changes.

Comment 1: This study on the parameters of a "new" beneficial predator is very well done. All parts = introduction, methods, results, and mainly discussion are well-written. Only the conclusions chapter is too long and rather based on literature than on own discoveries. Response 1: Thank you. I have reduced the discussion/conclusions.

Comment 2: line 143: Because of the functional response rules, you have to determine roughly the number of prey provided. Response 2: Yes, good comment. There were about 25-50 nymphs per day for the 1st and 2nd instars and about 50-75 per day for the 3rd instar. I have called this study 'prey consumption' and not a 'functional response' in part because we did not take exact counts of prey.

 Comment 3: line 186: Can you summarize lower thresholds and optimal temperatures in a table together with standard errors? Response 3: I don't believe the model I used had standard errors. I'd rather save space and not have another table for what is in the text just two sentences. I can make this change if required. On the manuscript, the reviewer made an good argument for the table, including "And that means whether they obey the developmental-rate isomorphy rule by Jarosik et al." I think this is beyond this simple study, but I have added the proportion of time spent in egg, larval and pupal stages, and added a sentence to the discussion. 

 Comment 4: line 206: Find a suitable non-linear equation for body mass. Response 4: This is a valid but tough ask. I did look up models that could fit the data best, but they had no biological connection to what the parameters were. I found models for body size and temperature, but these were quite complicated and often required more information on the tested animals than I had collected (especially mortality). Also, I am trying to keep the results on the biology presented as simply as possible to best compare with other articles and to reflect upon the later conclusions on insectary rearing. For that reason, I used a simple linear model because as temperature increased weight does go down. I have entered some text in the discussion to help explain this better. I have also included a simple quadratic equation that better fits the data set. This is a valid point raised by this reviewer and I hope that my changes are sufficient.  

PDF - The Reviewer caught some errors and made suggestions to a pdf. I have made all of the changes suggested. Primary to this is shortening the Discussion and Conclusion.